# Who supports Bernie? Analyzing identity and ideological variation on Twitter during the 2020 democratic primaries

**Stef M. Shuster[1]\*, Celeste Campos-Castillo[2], Navid Madani[3], Kenneth Joseph[3]**

**1** Lyman Briggs College and Department of Sociology, Michigan State University, East Lansing, MI, United States of America, **2** Department of Media & Information, Michigan State University, East Lansing, MI, United States of America, **3** Computer Science and Engineering Department, University at Buffalo, Buffalo, NY, United States of America

\* sshuster@msu.edu

**Data Availability Statement:** The data underlying the results presented in the study are available from https://github.com/kennyjoseph/bernie.

## Abstract

Using a novel dataset of 590M messages by 21M users, we present the first large-scale examination of the behavior of likely Bernie supporters on Twitter during the 2020 U.S. Democratic primaries and presidential election. We use these data to dispel empirically the notion of a unified, stereotypical Bernie supporter (e.g., the "Bernie Bro"). Instead, our work uncovers significant variation in the identities and ideologies of Bernie supporters who were active on Twitter. Our work makes three contributions to the literature on social media and social movements. Methodologically, we present a novel mixed methods approach to surface identity and ideological variation within a movement via use of patterns in who retweets whom (i.e. who retweets which other users) and who retweets what (i.e. who retweets which specific tweets). Substantively, documentation of these variations challenges a trend in the social movement literature to assume actors within a particular movement are unified in their ideology, identity, and values.

## Introduction

Social movements actors aim to bring about social change that aligns with their *ideology*, or their mental model of the world as it is and how it should be ([1], pg. 309). More specifically, an *ideology* is "a system of meaning that links assertions and theories about the nature of social life with values and norms relevant to promoting or resisting social change" ([2], pg. 43). A *value*, in turn, is a moral, ethical, or solidaristic commitment to some groups or social conditions as right or wrong, good or bad, moral or immoral, important or unimportant" ([2], pg. 43, see also [3]). To accomplish their goals, movement actors communicate the values inherent in their ideology and the social issues associated with them through *framing activities* [4]. These *framed values* and shared ideologies are loosely tied to a *collective identity*, or a shared label that social movement actors apply to themselves to characterize a shared sentiment about the movement [5]. All together, then, *social movement actors are oriented by their ideology, motivated by a collective identity, and communicate their values through the process of framing* [2, 6].

**Funding:** KJ and NM are supported by a grant from the Office of Naval Research (https://www.nre.navy.mil/) MURI ONR #N00014-20-S-F003 and the National Science Foundation (https://nsf.gov/) #IIS-2145051. The funders had no role in study design, data collection and analysis, decision to publish, or preparation of the manuscript. ss and CCC received no specific funding for this work. There was no additional external funding received for this study.

**Competing interests:** We have read the journal's policy and the authors of this manuscript have the following competing interests: ss, NM, CC, KJ are restricted in sharing complete replication data by the Twitter Terms of Service. Instead, data will be shared as described in the text. This does not alter our adherence to PLOS ONE policies on sharing data and materials.

Social movement actors in a particular movement are often perceived to share a single identity and ideology, and presented as such by mass media outlets [3]. However, what ultimately unifies actors within a movement is a common *goal* [7]. Multiple ideologies [2, 8] and identities [5] therefore can and do exist within a movement, especially when there is competition among a number of people who seek a nomination [9]. As pertaining to the present work, electoral campaigns, as a form of social movement, are often assumed to represent a homogeneous ideology and identity. This assumption incorrectly flattens the range of framed values, identities, and ideological perspectives actually held by campaign supporters.

Given its scale and diversity, social media presents a setting where we can look to explore variation within social movements, and specifically here, within an electoral campaign. Indeed, prior work has shown that variations in framing may be exposed [10] and studied [11] using social media data, as can variations in expressed identities as social movements arise and act [12]. Even in the expanse of social media, however, most existing work lumps users into broad categories, such as those aligned with particular candidates or political parties.

The present work addresses these gaps in the social movement and social media literatures through a case study on identity and ideological variation within the movement supporting Bernie Sanders' campaign for the Democratic presidential nomination in 2020. Using a novel dataset of 590M tweets from 21M users during February-December of 2020, we ask, **did supporters of Bernie Sanders on Twitter in 2020 exhibit identity and/or ideological variation, and if so, what did that variation look like?**

Popularized in 2015 by *Atlantic* writer Robinson Meyer as "Bernie Bros" [13], Bernie supporters online have largely been characterized as misogynistic, [14] racist, [15], and antagonistic white millennial men in popular media [16]. In line with this stereotype, hostility towards women and Black candidates and supporters was often on display, especially in media coverage of Bernie supporters [16]. At the same time, surveys suggest Bernie supporters were more likely to be young, Hispanic, Independent, and/or non-religious compared to other Democratic primary voters ([17], pg. 64). The Bernie campaign thus serves as an interesting case study of potential identity and/or ideological variation in a social movement because 1) the media and academic scholarship have coalesced around a narrative of a "typical Bernie supporter" (e.g., the "Bernie Bro") that 2) does not appear to match empirical realities of a more diverse base.

To uncover variation (if it exists), however, we must first address two difficult methodological challenges. First, **how do we identify Bernie supporting accounts?** Second, **how do we measure identity-based and ideological variation, or a lack thereof, within the content shared by these accounts?**

We employ a novel mixed methods approach to address these challenges. At the core of our work is the assumption, shared by researchers in Communications [18], Political Science [19], Sociology [20], and Computer Science [21]—that data on retweets constitutes a foundational relationship [22] for understanding user behavior on Twitter. Our work focuses on two ways of using retweet relationships. In the first, we focus on who retweets *whom*, creating a social network of retweeting behavior. We use this social network, and the profile descriptions of users, to identify distinct groups of users who are likely to support Bernie. Where multiple such groups exist, this analysis can also help us to understand variations in collective identity that emerge across groups. In the second, we focus on who retweets *what*. Prior work has largely focused on analyzing tweet content by clustering together tweets with similar text. Here, we instead analyze clusters of tweets retweeted by similar groups of people. This allows us to look at variation in the framed values shared by different collections of Bernie supporters in a new way.

Our analysis uncovers separate subcommunities of Bernie supporters that display quantitatively and qualitatively distinct social relationships, collective identities, values, frames, and ideologies. Specifically, we identify five distinct collective identities, ranging from the stereotypical Bernie Bro to committed socialists, who draw from three different ideologies: moderate, progressive, and far left to make the case for Bernie as president. Surprisingly, we further find that individuals with particular collective identities use framed values across the ideological spectrum, mixing and matching their content in search of the right message. As others have lamented, too often scholars overlook variation on these dimensions within movements [2, 9], which ultimately leads to overly simplistic accounts of how movements persuade audiences to support particular issues and/or candidates.

Collectively, our work makes three contributions to the literature on social media, social movements, and the methods used to study them:

1. Methodologically, we present a novel mixed methods approach to identifying identity and ideological variation, through the emergence of social groups with distinct identities and sets of retweets containing shared framed values symptomatic of distinct ideologies.

2. We present the first exploration of the community of Bernie supporters on Twitter. Documenting variation in identities and ideologies, we extend existing knowledge of social movements by dispelling empirically the notion of a unified, stereotypical Bernie supporter.

3. We provide a case study of how individuals with specific collective identities mix and match framed values from distinct ideologies over time. Identities and ideologies are intertwined [5], but political ideology and political identity are often treated as synonymous [8, 17]. We provide empirical evidence of the utility of distinguishing these two concepts.

## Additional background

Our substantive research questions are informed by 1) the sociology of social movements and 2) the study of social movements online, including work focusing specifically on the Bernie campaign. Methodologically, our work is informed by research on 3) the identification of users that share a common feature (here, a collective identity) and 4) content analysis of social media data. We briefly review the relevant research across these four literatures in separate subsections here.

**Social movements, framing, values, identity, and ideology.**   Social movements expend considerable time constructing "versions of reality, developing alternative visions of that reality, attempting to affect various audiences' interpretations, and managing the impressions people form about their movement" ([8], pg. 678). They seek to build credibility, persuade audiences to join the cause, and set the terms of political discourse about issues that matter to them [23]. Movements persuade audiences through framing activities. A *frame* is an interpretive schema used to help people understand a situation [24] by selectively "punctuating and encoding objects, situations, events, and experiences within one's present or past environment" ([25], pg. 137). Related, fram*ing* is the *process* through which reality is constructed by movement actors [25–27] and gives issues or events significance [28, 29].

Social movements frames can mobilize people, increase the likelihood of buy-in to the cause when grievances are presented in a way that resonates with people's values [30], and change behavior and opinion [3, 31]. While there is a significant body of scholarship on social movement framing, scholars [2] have advanced the concern that many studies disregard ideology as a distinct but important mechanism for movement buy-in, advancing the framed values and issues of a movement.

In political campaigns, ideologies filter frames by increasing resistance to disconfirming information [32] or providing reasoned principles to make sense of public debates [9]. For example, the "US healthcare system is killing people" frame may appeal more to a far left flank that supports distributive justice while "The US healthcare system is unjust" may appeal more to a moderate base [33]. Both ideological perspectives on health inequity are united by an overarching goal of improving social conditions and the healthcare system, but how to resolve health inequities and why it "should" matter to a populace will be filtered and framed differently through differing ideologies.

Differences in framed values are therefore symptomatic of differences in ideology. This is critical, as framed values are what is seen in the messaging produced by movement actors, e.g. in their tweets. Put another way, tweets are frames that communicate values that are shaped by ideological perspectives. We therefore study variation in ideological perspectives (e.g. moderate, progressive, or the far left) that shape framed grievances and/or values and identities by examining tweets.

In addition to using Twitter to express and frame values, movement actors also convey identity through the content they share [34]. Identity on Twitter is also known to be expressed on Twitter user profile descriptions, or bios [35]. While ideology and identity are analytically distinct, scholarship on American politics often conflates political identities and ideologies [3]. Recent work on ideology has shown, however, that ideological perspectives vary considerably across individuals with shared political identities [17], and this may be particularly true of politically active individuals [36]. Here, we seek to identify collective identities separately from ideology, and to then assess how these two overlap.

**Social movements, social media, and "Bernie Bros".** Scholars have long explored how social movements use, and mobilize on, social media (see, e.g., early work on the Arab Spring [37–39] and more comprehensive book-length treatments from Castells [40] and Jackson et al. [41]). Most relevant to our work, however, is scholarship that *uses social media data to understand movements*, rather than understanding how movements use social media. Specifically, we use Twitter data as a window into ideological variation among Bernie supporters. Others that have analyzed activity patterns of movements on Twitter have focused on the structure [42] and dynamics [43] of networks, the evolution of content from movements and associated counter-movements [44], the spatial [45] and demographic [46] variation of movements, and their relationship to offline discourse and activity [47]. More closely related is work exploring the emergence of particular collective identities over time in movements [48] and how framing strategies are appropriated on Twitter by movements [10]. We complement this prior literature by focusing explicitly on identity and ideological variation and how we can find such variation within supporters of a single movement.

More specifically, our analysis examines the online behaviors of the movement to secure Bernie Sanders' bid for the Democratic nomination during the 2020 presidential primaries. First popularizing the term Bernie Bros during the 2016 presidential election cycle, *Atlantic* writer Robinson Meyer [49] observed that the Bernie Bro is someone who "seems to have taken Sanders' rhetoric that America is trapped in a number of deep, unprecedented crises to heart and always writes with an urgent, anxious seriousness when discussing national politics." In jest, Meyer characterized Bernie supporters as an obstinate and earnest group. But the "Bernie Bros" phrase gained traction and transformed into a trope that portrayed *all* Bernie supporters as antagonistic misogynistic millennial racist males. In 2016, media pundits used the phrase "Bernie Bros" to describe a group of people who refused to accept that Hillary Clinton won the Democratic Party's nomination because the primaries were rigged [13], as well as disbelief that she won stemmed from deeply held misogyny [14]. Similarly in 2020, Bernie supporters were homogenized as Bernie Bros who refused to accept that Joe Biden won the nomination, as

Bernie held a considerable lead over other Democratic candidates until Super Tuesday, signaling another rigged election and conspiratorial thinking among so-called Bernie Bros [50, 51].

There are ongoing debates about how much the Bernie Bros trope accurately represents Bernie's supporters. Altamura and Oliver [17] present evidence to the contrary, showing that during the 2020 election cycle, Bernie supporters were more likely to be young, Hispanic, Independent, or non-religious compared to all other Democratic primary voters (pg. 644). They also find that younger cohorts were more likely to vote for Sanders *especially* if they supported government-sponsored health insurance and were low in racial resentment. At the same time, a litany of lived experiences present evidence that the "Bernie Bro" catechism accurately described behaviors of at least some Bernie supporters.

Our work compliments the empirical work from Altamura and Oliver [17]—to our knowledge, the only existing academic article to quantitatively analyze Bernie supporters—by shifting away from voting behaviors and examining the ideologies and identities of Bernie supporters online. In particular, we look to frequently retweeted content as a potent signal, noting that, "political elites and journalists construct issues in value terms as a heuristic for the broader public, which often understand social issues not in finite detail but in terms of the values they present" ([31], pg. 513)

**Analyzing collective identities in Twitter data.** The literature presents a number of different approaches to identify groups of users with particular characteristics or identities. The most straightforward is to use manually crafted heuristics based on user behavior, e.g. using whether or not users follow a pre-specified list of politicians to label those users as Democrats or Republicans [52], or using the sharing of specific hashtags to identify members of social movements [44].

One important question is thus what behaviors are used for identification. In the present work, we focus on 1) retweets and 2) profile descriptions. Specifically, we use the retweet network to identify groups of users who commonly retweet the same set of accounts, and profile descriptions to interpret collective identities shared within those clusters. Despite user claims that retweets "are not endorsements," survey data suggests that retweets predominantly serve to promote or re-up tweets [53]. Because of this, prior work has shown that retweets can readily identify distinct social groups [21], and that analyzing profile descriptions of these users can uncover shared identities within the group [18].

Regardless of what behavior is used, however, the use of manual heuristics has known limitations. One such limitation is that manual heuristics provide a high precision, low recall identification strategy. For example, individuals who follow multiple Republican politicians are likely to be Republicans, but few Republicans follow right-leaning politicians [54]. One way to deal with this issue is to apply supervised machine learning methods, where manual heuristics serve as the training data [52]. Here, we adopt a machine learning approach to avoid this problem of low recall, but use a method that also seeks to address a second limitation to manual heuristics: relying on a predefined heuristic potentially limits us to a subset of the relevant users. In our setting, for example, predefined heuristics to identify Bernie supporters might lead us to miss ideological diversity if our preconceived notions of Bernie supporters limits us to a subset of the overall population.

Specifically, we adopt the recently proposed approach of Zhang et al. [18], who developed a mixed methods paradigm to identify what they call "flocks" based on patterns in who retweets whom. Flocks consist of sets of users and the accounts they commonly retweet. Flocks are first identified via a computational technique known as Vintage Sparse PCA, described in more detail below. Each flock represents a set of influential users and the users that commonly retweeted them. Social identities collectively shared by the group can then be identified, if they exist, via analysis of user profile descriptions and tweets.

**Analyzing values and their frames in Twitter data.** Social media researchers have developed a number of strategies to identify the values and frames expressed by social movement actors. Qualitative approaches are generally restricted to a small subset of the data, but provide nuanced insights. In contrast, purely quantitative approaches can scale to tens of millions of tweets and thus identify, if only bluntly, rare behaviors that would escape study in a small sample.

Recent work has adopted a balance between these two approaches. These mixed methodological approaches include, for example, active learning methods that adopt models to a scholar's qualitative coding as the codes evolve [55], and the use of unsupervised machine learning methods to filter data into clusters or groups which are then analyzed qualitatively [56]. The present work takes the latter approach. While some have advocated for iterative use of unsupervised machine learning methods and qualitative analysis [57], we here find that a single application of a machine learning model and subsequent qualitative analysis is sufficient for our research question.

Our work also differs from previous research in what data we apply a machine learning model to. Prior research focuses on the application of machine learning to cluster social media posts into themes, or more specifically to cluster posts based on their content (and potentially, associated metadata) [58]. In contrast, the present work creates clusters of tweets based on *who retweets what content*. That is, we identify groups of tweets that are retweeted by similar sets of users. We discuss in more detail below the motivations for doing so.

## Materials and methods

Data collection was performed using the *tweepy* [59] library; code for collecting and storing data have been released as part of the code release for this article. Data were accessed in ways that comply with the Twitter API Terms of Service. Our methodological approach consisted of two primary steps: a method to identify Bernie supporters and variations in collective identities, and a method to identify variation in the framed values and ideologies espoused by these Bernie supporting accounts. To identify Bernie supporters and identity variants within groups of Bernie supporters, we adopt and extend the methodology proposed by Zhang et al. [18], relying on the "who retweets whom" network to extract clusters of influencers and ordinary users. We define *influencers* as accounts that are commonly retweeted in the dataset and thus serve influential roles in communication about the 2020 election. We define *ordinary users* as all other users in the dataset. While we adopt the overarching strategy proposed by Zhang et al. [18] for identifying clusters, we extend it by 1) providing a new method to select the number of clusters, $k$, and 2) proposing new quantitative evaluation metrics to help validate qualitative identification of relevant clusters. We then use a similar approach, but on the "who retweets what" network, to extract clusters of specific tweets retweeted by similar groups of users. We use these sets of tweets to assess variations in ideology. To the best of our knowledge, no prior work has explored patterns that focus on clusters that emerge from the who retweets what network. These methods used the collected data in ways that comply with the Twitter API Terms of Service.

Replication data and code for this work can be accessed at https://github.com/kennyjoseph/bernie. Replication data is released in accordance with Twitter's Terms of Service; as such, raw data used in the study cannot be made publicly available. However, following recent guidance from computational social scientists [60] relevant to the challenges with replicability under Twitter's current API settings, we will make the raw data available to researchers upon request to the last author.

## Data

From January 27th through December 26th, 2020, we used the Twitter V1 Streaming API to track tweets containing keywords or user mentions relevant to the 2020 Democratic primaries and the 2020 U.S. national election. Our keyword list was updated manually as new events arose; for example, we added several keywords as hashtags arose around various primary and/ or presidential debates. Keywords and accounts tracked reflect a broader interest in the 2020 election, and thus contains keywords likely to be used across the political spectrum. Over the course of data collection, we tracked a total of 229 keywords and mentions of 20 different accounts. Lists of both of these are provided here in the Supporting Information.

Decisions on how to select keywords to identify relevant tweets for a specific topic can have a critical impact on our understanding of particular events [61]. Given the importance of these decisions, scholars have developed a number of automated methods to inform keyword selection [62, 63]. In contrast to the manual keyword selection approach used here, these automated methods have been shown to improve recall, in that they are able to collect more potentially relevant tweets [63]. As with other efforts favoring precision over recall [64], our use of a manual keyword approach is, therefore, more conservative. Doing so allows us to be more confident that any ideological variation we observe across Bernie-supporting accounts in our dataset is associated with relevant political discourse, as opposed to discussion of other unrelated events. Moreover, we follow a similar goal in the manual keyword selection process, opting to include keywords only after exploring their use with the search function on Twitter's website and confirming that the majority of tweets related to the hashtag appear to be about the election.

In total, our data consists of 589,729,860 tweets sent by 20,928,178 users. Fig 1 shows the temporal distribution of our data and reveals that our data is largely centered around tweets about Super Tuesday and then the lead up to the Democratic National Convention through the end of the general election. The figure also reveals a number of gaps in our collection in the period of May through July, where power disruptions at our university led to unanticipated and not immediately diagnosed errors in our collection.

Of the nearly 590M tweets in the dataset, only 169,614,594 (28%) were original content; the other 72% of tweets were retweets. Moreover, a large portion of retweets were concentrated in

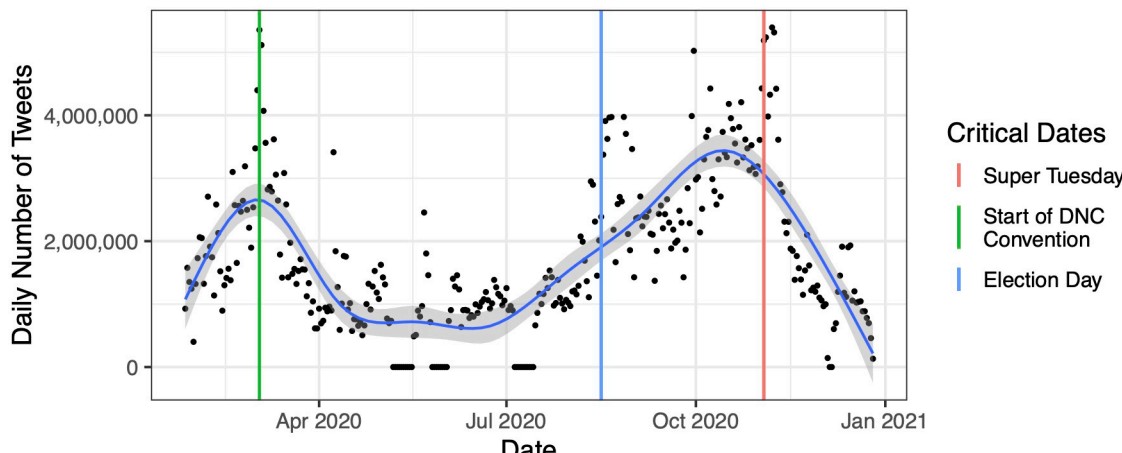

**Fig 1. The number of tweets (y-axis) per day (x-axis) in our overall dataset.** Each day is shown as a point, with a smoothed line (using a generalized additive model with cubic splines and 10 knots) showing the moving average and a 95% confidence interval around it. As described in the legend, the first, vertical, orange line is the date of Super Tuesday, the middle, purple line is the start date of the Democratic National Convention, and the last, green line is the date of the general election.

a small number of highly-retweeted tweets. For example, 4355 tweets were retweeted more than 10,000 times. Despite these 4,355 tweets themselves accounting for only 0.003% of the data, retweets of these tweets accounted for nearly 15% of the full dataset. Retweets, and in particular retweets of a small set of very popular tweets, thus constitute the majority of our data.

## Identifying Bernie supporters and identity variation

**Constructing the "who retweets whom" matrix.** We began by creating a matrix of who retweets whom in our sample, where ordinary users who retweeted others are the rows of the matrix, and influencers who were frequently retweeted are the columns. Overall in our data, 14,244,949 users (68% of all users) sent at least one retweet, and 2,308,610 users (11%) were retweeted at least once. The who retweeted whom network contains 166,021,415 total links.

Aligned with Zhang et al. [18], we performed a filtering step on this larger matrix. Critically, this biases our analysis towards heavier users of Twitter within our dataset. The core focus of our work, however, is on understanding whether or not there exists *any* variability. Consequently, finding variation in the sample of users we study is at worst an underestimate of the overall variability of Bernie supporters on Twitter. Our filtering step retains ordinary users who retweeted at least 12 different users, and influencers retweeted by at least 500 different users. These filter levels were selected by identifying the lowest levels at which 1) we observed interpretable clusters (as in Zhang et al. [18], we found that very low levels of filtering resulted in significant noise in cluster results), and 2) that were computationally feasible on the hardware available to us and likely available to other academic researchers (in our case, a single server with 16 CPU cores and 64 gigabytes of memory). This filtering step limits our analysis to 1,791,912 ordinary users, or 13% of those who sent at least one retweet, and 25,395 influencers, or 1.1% of all accounts that were retweeted at least once. Importantly, however, this sample retains 67% of the original links in the who retweets whom network, and accounts for over half (325M, or 55%) of all tweets in our dataset. Thus, while we sampled a small subset of users in our dataset, we retain the majority of content.

**Finding clusters of users using vintage sparse PCA.** Following Zhang et al. [18], we then use Vintage Sparse PCA (VSP) to identify groups of ordinary users and the influencers that they commonly retweet. VSP performs the same computation as the traditional factor analysis in psychology: a principle component analysis followed by a varimax rotation. As such, both ordinary users and influencers can be said to "load onto" particular latent dimensions. We can then use the output of the factor analysis to place both ordinary users and influencers into groups by placing them into the factor they load the highest on [65], where these loadings onto each dimension are called *factor loadings*.

Our choice of VSP is motivated by a number of factors. First and foremost, it is feasible to run VSP on our data. Put another way, VSP is scalable; runs of VSP on matrices that involve millions of rows and tens of thousands of columns, as in the present work, can be carried out in under ten minutes. Relative to unimodal clustering algorithms (e.g. the Louvain methods [66]), VSP provides a principled approach to identifying the most influential accounts in each cluster, by simply looking at factor loadings of the matrix columns (influencers). Relative to other scalable, bi-modal clustering algorithms used in prior work analyzing Twitter data [67], VSP provides stronger theoretical (in a statistical theory sense) guarantees on robustness to non-normal inputs, and produces a sparse representation that in our initial analyses presented more salient differences across clusters.

Relative to other scalable, sparse bi-modal algorithms widely used in the literature, VSP has close links to other widely studied models. In particular, when data are not centered before applying the method, VSP is mathematically equivalent to the stochastic block model, a widely

used hard clustering model for social networks [68]. Because we desire a hard clustering (i.e. every user is assigned to one and only one cluster), we use this approach to cluster the "who retweets whom" network. When data are centered, applying VSP is mathematically equivalent to latent Dirichlet allocation [69], a widely-used tool for soft clustering (i.e. Bayesian ad-mixture modeling) in the computational social sciences. We adopt this approach to cluster the "who retweets what" network, where we anticipate that users adopt multiple framed values instead of being fixed to just a single approach. Finally, VSP has been proven in prior work to be effective on the similar task of clustering the follower network on Twitter [18].

VSP thus presents an attractive tool for our work. However, as with any tool, there are notable limitations. First, VSP does not provide clear guidance on whether or not to scale inputs before use [65]. We opt not to scale in the present work, meaning our results may be more sensitive to outliers. Second, unlike some methods, VSP requires us to select the number of latent dimensions (or equivalently here, groups) to model. Recent work has argued that commonly used approaches to identifying the "correct" number of latent factors in a factor analysis may discard dimensions that hold potentially valuable data [70]. As such, we treated the problem of determining how many latent dimensions (clusters) to select as a task of finding a setting of $k$ that is relatively consistent with other potential settings. To do so, we followed the method of Joseph et al. [67]. We first ran VSP with a variety of values for $k$, iterating from 5 to 60 in increments of 5. We then compared how consistent (in terms of the Adjusted Mutual Information [71]) the clustering produced by each value of $k$ is with all other clusterings. We find that AMI is maximized when $k = 35$, and as such we selected that as the final $k$. See the Supporting information for additional details on the selection of $k$.

**A mixed methods approach to identifying Bernie supporting clusters.**   Given the output of a particular VSP model, we propose a multi-step, qualitative approach to identifying Bernie supporting accounts, as well as identity variations within those accounts. As a first step, we generated a sample of the top 25 influencers and top 25 ordinary accounts for each cluster. We defined "top 25" in each case using the factor loadings, which are, informally, a numeric value that represent the VSP model's estimate for how representative a given influencer (ordinary account) is of a given cluster. We then conducted a first round of open coding [72] with information about these users, including their profile descriptions, usernames, locations, etc. Three paper authors then independently provided a label for the cluster and a decision as to whether or not the cluster appeared to represent Bernie supporters. In a second step of focused coding [72], for any cluster that at least one of the authors believed was a group of Bernie supporters, all authors additionally looked at the top 50 retweets for that cluster in terms of overall shares by ordinary users within the cluster. The three authors then came to a consensus decision on whether or not each of the 35 clusters represented Bernie supporters or not.

Once final decisions were made, we conducted a pair of quantitative evaluations to assess evidence from our qualitative findings, which were restricted to only top users, across all of the users within the clusters. Our first assessment focused on the extent to which ordinary users in the cluster were likely to retweet Bernie Sanders relative to the other candidates in the 2020 Democratic primaries, or Donald Trump. We measure the likelihood of retweeting one candidate relative to the others by computing the (weighted) log odds of a user within the group retweeting each candidate relative to all others, using the formula in Equation 21 of [73] and as implemented in the R package *tidylo*. For our second quantitative evaluation, we first identified a set of character strings we found to be representative of Bernie supporting accounts in our qualitative work. We then scanned profile descriptions of all accounts and determined the percentage of influencer and ordinary accounts in each cluster that contained the character strings "bernie," "sanders," "notmeus," "socialist," "Chapo," or the red rose icon, which implies a socialist political orientation.

Finally, we use insights from both our qualitative and quantitative analysis to explore how collective identities varied across the five social groups we found to contain users that were likely to support Bernie. In addition to these existing methods, we additionally computed the weighted log odds of users expressing particular phrases in their profile descriptions, following the approach of Pathak et al. [35].

## Identifying ideological variation of Bernie supporting accounts

We used a similar process to the one described above to identify ideological variation. The core difference is that instead of focusing on who retweets whom, we focused instead on who retweets which specific tweets. To begin, we subset our analysis to the 213,645 users who 1) were in a cluster of the who retweets whom network that we labeled as Bernie supporters, and 2) sent at least 10 retweets. We then subset our analysis to the 21,901 tweets that were retweeted by at least 100 distinct users. With this dataset, we repeat our AMI-based procedure to identify an appropriate setting of $k$, ultimately settling on 20 (see the Supporting information). Following Zhang et al. [18], we analyze here only the 14 clusters that were associated with more than 1000 user accounts. As in the prior work, we make this decision in order to avoid emphasizing themes that emerged from a minority of users and may thus not reflect a widespread value or ideological perspective. Tweets that are in the same cluster after this procedure thus represent tweets shared by similar subsets of users, and users who are in the same cluster represent those who retweeted those tweets. Empirically, we should thus expect that tweets in the same cluster represent a shared set of framed values that individual users see as intertwined.

Our qualitative analysis uses the same two-stage approach to coding, first open coding each cluster based on content (i.e. framed issues and values conveyed). We then ascribed a label for each cluster (e.g., "Character attacks"). In a second round of coding, we used focused coding for ideology, based on extant literature distinguishing between moderate, progressive, and far left ideologies. Each of the steps in our qualitative analytic techniques emphasized alignment with best practices in the social sciences and consensus among the team and a recursive process of coding the data based on themes that emerged and in reference to existing literature (see also [74]). Ultimately, the qualitative analysis enabled us to identify the ideological variations that emerged in the framed values and issues each cluster represents.

## Results

In the following subsections, we first discuss results of our analysis of the who retweets whom network, where we identify five different social groups of likely Bernie supporting accounts with distinct identities. We then discuss results of our analysis of the who retweets what network, where we identify fourteen different clusters of framed values and the users that espouse them. These clusters, we argue, show evidence of three distinct ideologies—moderates, progressives, and the far left—that emerge from these sets of framed values in the who retweets what network. Finally, we show how these identities and ideologies interleave.

### Finding Bernie supporting accounts and identity variations amongst them

We find five distinct clusters in the who retweets whom network that represent social groups who are likely to be Bernie supporters: Groups 3, 5, 11, 19, and 27. A high-level overview of all groups is provided in Table S2, where we show the top 10 influencers (in terms of factor loadings) for each group. Groups ranged in size from around 5,000 users (Group 3) to almost 70,000 users (Group 5). Groups 11, 19, and 27 contained roughly 45,000 users, 25,000 users, and 65,000 users, respectively. We identify users in these groups as being likely to support

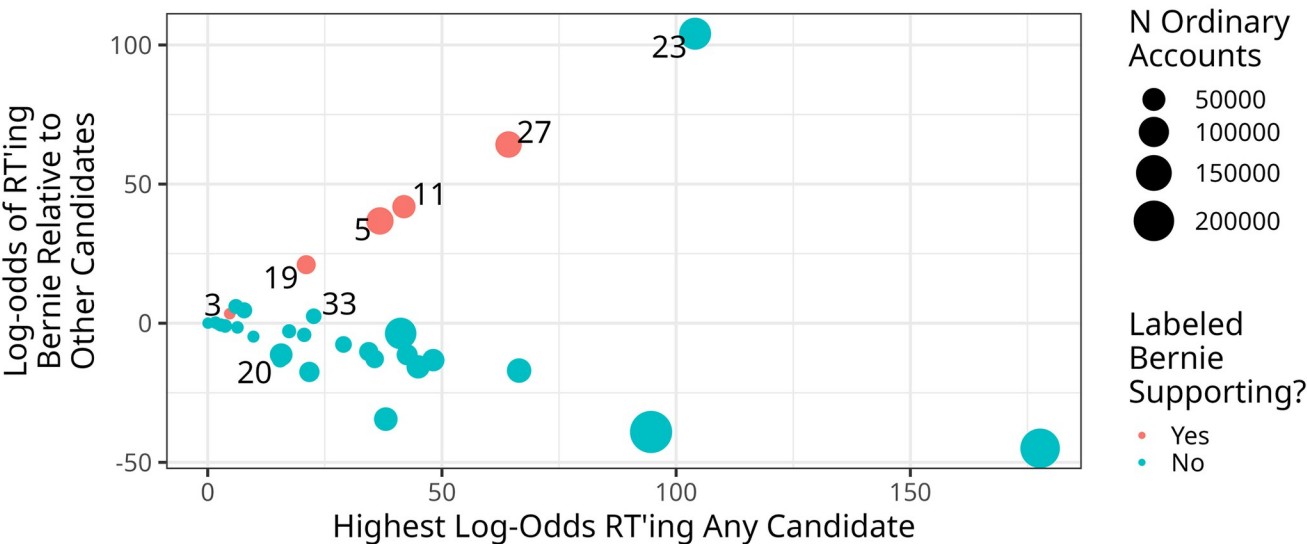

**Fig 2. For each cluster (each dot), the y-axis depicts the weighted log odds of accounts in the cluster retweeting Bernie relative to any of the other candidates in the 2020 Democratic primary or the 2020 general election (Joe Biden, Tulsi Gabbard, Elizabeth Warren, Andrew Yang, Pete Buttigieg, Amy Klobuchar, Michael Bloomberg, Donald Trump, Tom Perez, John Delaney, Tom Steyer, or Deval Patrick).** The y-axis represents the weighted log-odds of the most frequently retweeted candidate for that cluster. Where Bernie was the most frequently retweeted candidate, the x- and y-axes are therefore the same value. Points are colored by whether or not we labeled the given cluster as being likely to contain Bernie supporting accounts, and sized by the number of ordinary users in the cluster.

Bernie through qualitative analysis, but our quantitative validation checks provide some additional validation. Along with our qualitative work, they also offer insight into the variations in identity that existed across groups.

Fig 2 shows that with the exception of Group 3, users we identified as Bernie-supporting were more likely to retweet Bernie relative to other 2020 candidates. For example, Group 27 retweeted Bernie the most of any candidate, retweeting him nearly 60 times as frequently as anyone else. Fig 2 also presents two important caveats to our analysis. First, Group 23 was not identified as being likely to contain Bernie supporting accounts, but users in the group were 100 times more likely to retweeted Bernie than any other candidate. Our qualitative analysis suggested that while many users in Group 23 seemed likely to support Bernie, the primary focus of attention of these users was on the Black Lives Matter movement, and not the election itself. We therefore opted to avoid labeling the entire group as being likely to support Bernie. Such a decision is conservative, in that bringing these users into our analysis would almost certainly serve to expand variation in the identities and ideologies we uncovered. Second, users in Group 3 were only slightly more likely to retweet Bernie, suggesting that perhaps these users were not supporters of his candidacy.

Analyses of the profile descriptions of Group 3 users do, however, suggest that these users were likely Bernie supporters, and moreover provides little evidence of Bernie support from users in Group 23. Fig 3 shows that of the six groups with the highest average percentage of Bernie support-related keywords in their profiles, five are the groups we identified as Bernie supporters in our qualitative work. In particular, 40% of all ordinary users in Group 3 had a Bernie-supporting keyword in their profile. Group 19 had the highest percentage of influencers with a Bernie-supporting keyword, with almost half signaling one of these identities. Finally, an analysis of the two clusters with high levels of Bernie-supporting phrases that we did not label Bernie supporting, Groups 20 and 33, revealed a significant number of profiles with phrases like "never bernie" or "not bernie." This explains why these groups, which we

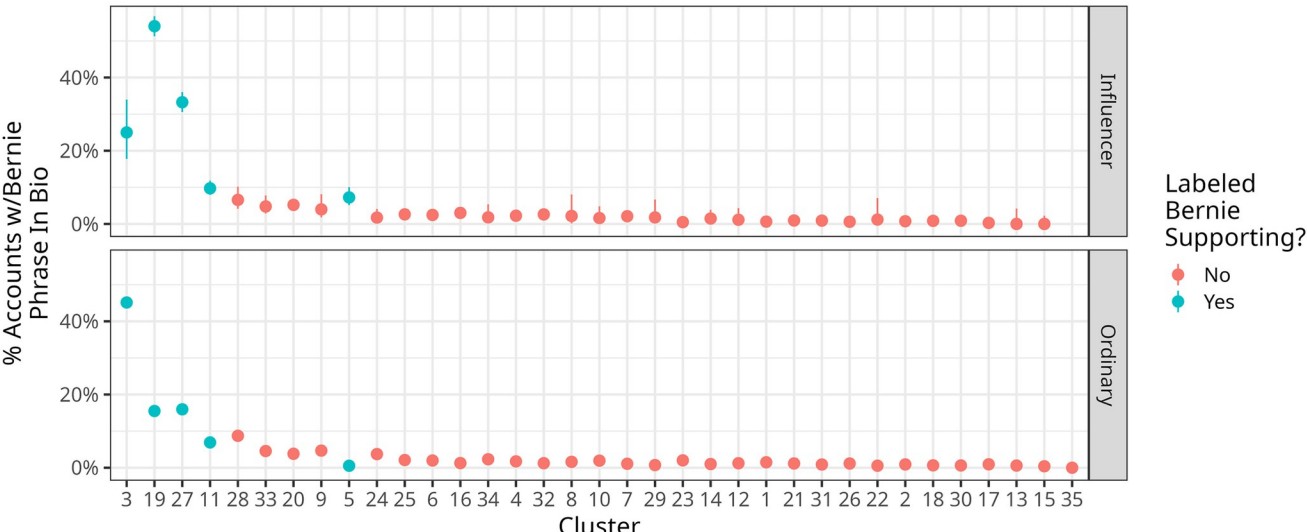

**Fig 3. The percentage (y-axis) of ordinary (triangles) and influential (circles) accounts in each cluster (x-axis) that contained in their profile description one of the Bernie-supporting keywords we identified in our qualitative analysis.** Points are colored by whether or not we labeled the given cluster as being likely to contain Bernie supporting accounts, error bars are 95% binomial confidence intervals.

labeled in our qualitative coding as Kamala supporters and Tulsi supporters, respectively, have high percentages of Bernie-supporting phrases in Fig 3 but are not Bernie-supporting accounts.

In summary, our mixed methods approach, along with our validation of it, reveals evidence of five distinct Bernie supporting groups. We now describe each of these groups in turn, highlighting distinctions in their within-group collective identities. Additional variation can be seen in the top 25 personal identifiers (in terms of weighted log odds) for each group; see the Supporting information for these.

*Group 3: Popular progressives.* Influencers in Group 3 included several progressive politicians, most notably Representatives Alexandria Ocasio-Cortez and Ilhan Omar, and Bernie Sanders himself. It also included more mainstream, and sometimes controversial, progressive activists such as Shaun King and David Sirota. As noted above, ordinary users in this group were most likely to have profiles signaling support of Bernie. Group 3 thus represented a more "mainstream progressive" identity, centering on open identity-based displays of support from ordinary users coupled with attention to well-known progressive figures.

*Group 5: Young and/or Black voices.* Both influencers and ordinary users in Group 5 frequently noted their ages in their profile descriptions; three of the top 25 phrases in the profile descriptions of influencers were age references of 18, 21, and 22 (see Supporting information). Group 5 users also frequently referenced Black identities, and presented both progressive (e.g. Black Lives Matter) and far left (e.g. "actual communist") ideologies in their profiles.

*Group 11: Bernie Bros.* Influencers in Group 11 included the co-hosts of the podcast *Chapo Trap House*, often cited as the catalyst for Bernie Bros. Influencers also frequently referenced other far left podcasts, such as the *Trash Future* podcast and *Eat the Rich*, as well as a number of print media outlets ranging from moderate (e.g. the NY Times) to far left (The Baffler Magazine).

*Group 19: Green Party Socialists.* Both influencers and ordinary users in Group 19 referenced identities relevant to socialist political identities: the most popular personal identifiers in the cluster included references to progressive parties including the People's Party, Green Party,

and #bernietogreen and #ventura2020, which reflects a transition to the Green Party's 2020 nominee for president once Bernie dropped out of the race. Profile descriptions of users also referenced several far left positions on issues, including #freecollege and #ClassWar2020.

*Group 27: Organizing for Bernie.* Influencers in this group included leaders in the Democratic Socialist and Working Families Parties, as well as a number of union leaders and mainstream progressive accounts that covered the Sanders campaign. Nearly half of the influencers in this group presented a Bernie-supporting progressive identity in their profile: 30% used the phrase "Bernie" or "#notmeus", 25% mentioned medicare for all, and almost 20% referenced socialism. Many of these profiles emphasized a call to action in support of a Bernie candidacy.

## Ideological variation of Bernie supporters

Despite research that tends to present supporters of a political candidate as ideologically unified [2] we identified fourteen clusters in the who retweets whom network that represent a spectrum of variation in ideologies of Bernie supporters. These ideologies in turn differentially shaped how they framed their grievances or issues. These variations unfolded over time in important ways; Fig 4 visualizes this temporal distribution. First, we note that across all ideologies and framed value clusters there was a lot of activity in the months leading up to Super Tuesday (March 2020). However, around the time of the Democratic National Convention (August 2020), framed value clusters reflecting progressive and far left ideologies ramped up.

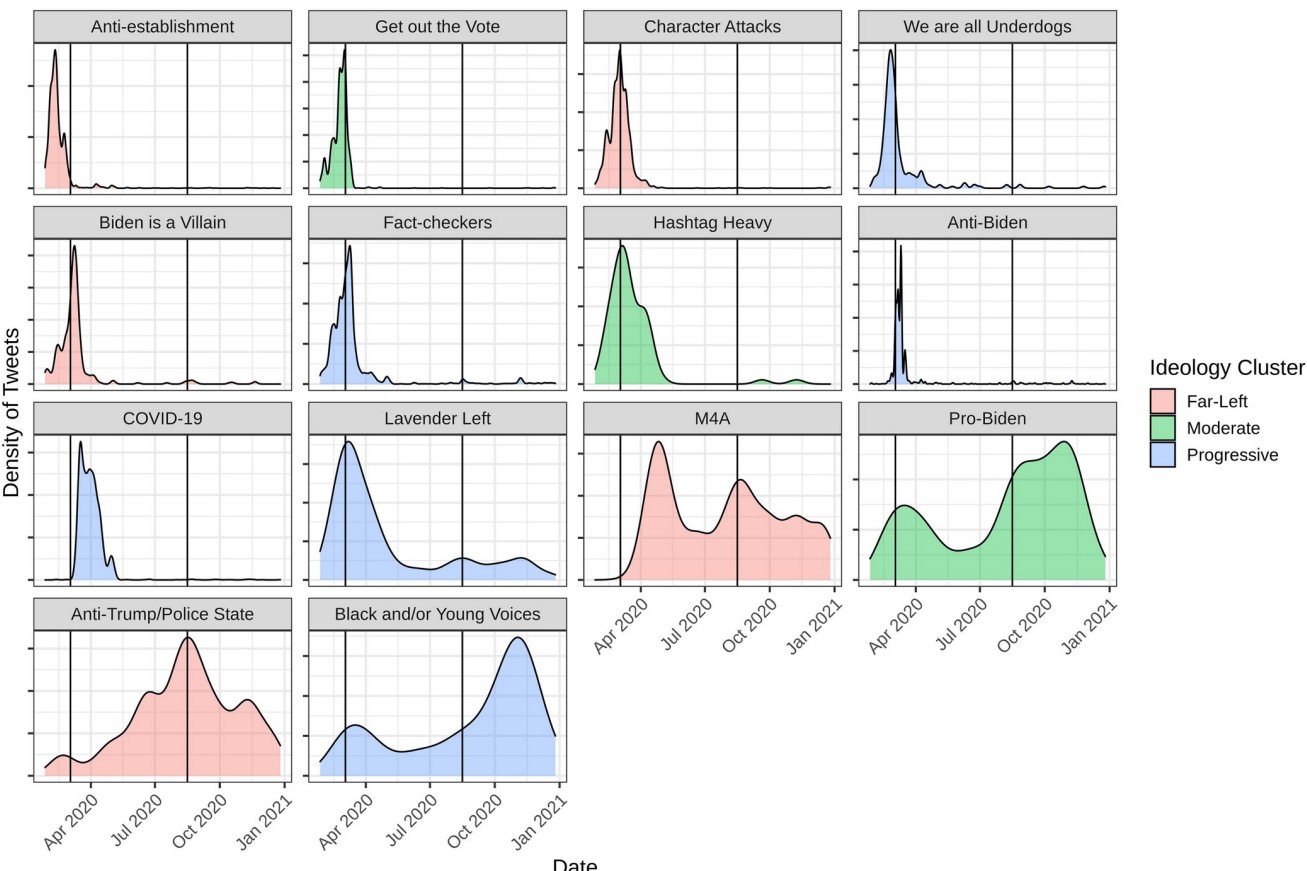

**Fig 4. The density of tweets (y-axis) over time (x-axis) for each cluster (separate subplots) of tweets we identified in the who retweets what network.** Clusters are colored by the ideology we associate them with in our qualitative analysis.

While one of Bernie's key progressive signature issue (Medicare for All) was represented across the election season, we note that anti-establishment values around the police state, the election itself, and Trump peaked during and after the DNC. Once Biden's nomination was secured in August of 2020, moderate framed values picked up; mostly offering endorsements of Biden's presidential bid. We organize our analysis below around moderate, progressive, and far left ideologies, and discuss each cluster of framed values that fit these ideologies (we recognize that these terms are dynamic, operationalized differently by different disciplines, and are historically contingent (see [33]).

**Moderate ideology clusters.** Tweets with framed values shaped by a moderate ideology were rare relative to those drawing from a far-left or progressive ideology, but still appeared to support Bernie during the 2020 election season. Drawing on Carmines and Berkman [75] we define a moderate ideology as one containing centrist views, support for the status quo, and instrumentally oriented. To this end, clusters of framed values we identified as emblematic of a moderate ideology were non-confrontational, passive conveyances of support, and friendly reminders to get out and vote.

*Cluster 13: Pro-Biden.* As mentioned above, a cluster of Bernie supporters signal boosted Joe Biden's bid to become president in the fall of 2020. This translated to re-tweeting Biden's campaign message that asked people to consider what would happen if Trump was re-elected. For some, the stakes were too high, even if Bernie's supporters were not convinced that Biden was the best Democratic candidate. As one of Biden's tweets rhetorically asked, "When Donald Trump says tonight you won't be safe in Joe Biden's America, look around and ask yourself: How safe do you feel in Donald Trump's America?" Almost all tweets presented here have been retweeted at least 100 times, often substantially more, are from public figures, and are still available via the API. Tweets following this one are thus quoted verbatim. In the few cases where the tweet is no longer available, we do slightly modify content to preserve privacy and do not provide a citation.

*Cluster 14: Hashtag heavy.* Tweets in this cluster relied heavily upon hashtags to demonstrate their support for Bernie and his platform of signature issues; signaling support by writing, "#MedicareForAll," "#BernieBeatsTrump." "#Bernie2020," "#NotMeUs" which was followed or preceded by the message of "Vote for Bernie." Hashtags symbolically convey a person's stance and are a mechanism for advocating for one's position [76]. We coded this cluster as 'moderate because most of the hashtags passively named issues or endorsement of Bernie, thereby upholding civility in discussions of politics [77].

*Cluster 19: Get out the vote.* Similar to the other moderate re-tweet clusters, this cluster's main concerns were instrumentally oriented in reminding people to get out the vote in the primaries, reach out to friends and family members to vote, and contribute money to Bernies' campaign which offered hope that a better future was possible if he was elected as the Democratic nominee, "Despite what the political establishment wants you to think, running a campaign of, by, and for the people isn't a weakness—it's our greatest strength. Together, we can prove it tomorrow"

**Progressive ideology clusters.** Whereas clusters of framed values representing a moderate ideology were likely to uphold the status quo, retweet election strategies, and signal passive support for Bernie, progressives were more likely to retweet grievances around social issues, inequities, or disdain for other candidates in the race. We draw on McCright and Dunlap [7] to define a progressive ideology in the most generalized sense as one oriented toward improving social conditions for various social groups or issues.

*Cluster 2: Anti-Biden.* In the days leading up to Super Tuesday, Bernie supporters became more vocal in stating their opposition to Biden and calling attention to Biden's historical record of conservative policy-making such as supporting the Iraq War, cutting social security,

and the Defense of Marriage Act (DOMA). As one prominent retweet stated, "The American people are sick and tired of endless wars in the Middle East which have cost hundreds of thousands of lives and trillions of dollars. We will not defeat Trump with a candidate like Joe Biden who supported and voted for the Iraq War alongside Republicans."Tweets in this cluster were concerned that Biden's centrist policies and policy record appeared to be aligned with Republican values.

*Cluster 6: Young and/or Black voices*. While Bernie was often presented in the media as not being popular among Black voters, there was a cluster of Black users who vocalized support of Bernie or discontent with Biden. This was often, though not always, vocalized using humor. Humor in political mobilization can be an effective way to frame grievances or offer support [78] and this cluster used it in both ways to promote Bernie or to call out Biden and Harris for not being responsive to progressive Black voters. For example, as one retweet jokingly referenced Biden and Harris walking out of a building with a pole in between them, "They split the pole, we done for."And another reflected, "This poor guy is gonna die tryna save the U.S. and you are not paying him any mind." Voices representing "young people" also emerged here, with several tweets appealing to young adults to vote or to think specifically about youth and their life situations.

*Cluster 7: The Lavender Left*. Bernie had a consistent record supporting queer rights which this cluster emphasized, while calling out other candidates, such as Warren and Biden, who had previously voted in favor of anti-LGBTQ+ issues. As one tweet emphatically stated, "elizabeth warren used to be anti-lgbt and a republican. biden also was until recently. Here's our guy Bernie: [link]" Undermining the credibility of other candidates by positioning not only their policies but identities as anti-LGBTQ+, these frames discredited Biden, Warren, and their supporters for not upholding progressive values. Moreover, this cluster advocated for an interdisciplinary platform of gender justice and anti-racist progressive feminist platform while rallying against Biden's spotty and moderate neoliberal track record on abortion, women's equality, and the rights of sexual assault survivors. This frame extended to a critique of those who claimed to be progressive or liberal while supporting candidates who did not uphold these values, especially in the context of rape allegations against Joe Biden and promoting nationalism over progressive values. As one tweet suggested, ". . .I smile serenely at the lack of ideology."

*Cluster 9: Covid-19*. Faced with the Covid-19 pandemic during the height of the 2020 primary season, Bernie supporters began to make connections between one of Sanders' signature issues—Medicare For All—and the unfolding global health crisis. From a video link tweeted out on April 3, 2020, Bernie opined, "If there's anything the American people are learning during this horrific crisis is how weak and how dysfunctional our current healthcare system is. . ."

As one of the first candidates to begin speaking openly about the COVID-19 pandemic and potential economic hardship facing the country, Bernie framed his Progressive healthcare policies and hope for universal health coverage as a viable option for ensuring that all people had access to healthcare, even during a global pandemic.

*Cluster 15: Fact-checkers*. Scholars have consistently found that merely fact-checking potential mis- or disinformation does little to change people's perspectives (Hoffman 2015). Despite the lack of efficacy, one cluster used fact-checking tactics to draw attention to the inconsistencies in Bernie's opponents' voting record on issues that progressives typically support. Tweets in this cluster were similar to those in the Anti-Biden one (Cluster 2), but appealed to progressives by using 'fact-checking'-like language to position opponents as too centrist, or to frame Bernie as clearly on the right side of political debates in putting people before profits.

*Cluster 20: We are all underdogs*. The underdog is a powerful trope used in social movement mobilization to garner sympathy for one's position or grievance [79]. Progressives used the underdog rhetoric to position Bernie as a hero and champion for the masses. As one tweet

read, "Thank you Senator @BernieSanders for your compassion in these trying times, and for your solidarity with millions of laid-off workers, including tens of thousands @unitehere members in the hospitality industry. #ThankYouBernie." Bernie's policies and consistent concern for worker's rights, affordable housing, and other progressive policies not only offered hope for people experiencing inequalities, but enabled them to frame their concerns from an underdog perspective and Bernie as the person who was well-poised to become a powerful advocate and champion.

**Far left ideology clusters.** In contrast to clusters representing moderate or progressive ideologies, there were several clusters shaped by a far left ideology, which we define as one that upholds anti-establishment views and eschews institutional norms [51]. While the issues discussed were similar to those we labeled progressive, how these issues were framed was more emphatically conveyed, antagonistic, and hostile. At times, some of the clusters in the far left offered conspiratorial rhetoric to frame their grievances.

*Cluster 10: Character attacks on other candidates.* Engaging in character attacks against other candidates can be a potent strategy for discrediting an opponent and framing grievances [80]. One cluster engaged in such attacks, "If @JoeBiden was so good at building a coalition why did he need Klobuchar, Buttigieg, and Bloomberg to drop out and endorse him in order to get into a statistical tie with @BernieSanders? Because Biden is a bad candidate and Bernie's message speaks to more voters."

Using conspiracy-adjacent rhetoric, this cluster framed Biden's weakness as not only a failure of the candidate, but of the DNC in gatekeeping Bernie from proceeding to the top of the Democratic ticket. While clusters from moderate and progressive ideologies framed opposition to Biden and support for Bernie, a far left ideology attacked Biden's character by calling into question his integrity as a nominee and ability to win an election, which was further supported by the perceived coup by the DNC.

*Cluster 11: Anti-Trump/police state.* A distinct group of Bernie supporters were gravely concerned that Trump was attempting to suppress democracy, "Make no mistake about it. This is a blatant attempt by Trump's handpicked Postmaster General (and campaign contributor) to sabotage the Postal Service, suppress the vote and undermine democracy" thereby making it difficult for mail-in ballots to reach their intended destination while turning cities into a police state, "Against the wishes of state and local officials, unmarked federal agents in combat gear have invaded parts of Portland, Oregon. This is what a police state looks like. These troops must be withdrawn immediately. We must defend the Constitution. We must defeat Trump." Noting that Trump was acting against the wishes of voters, elected officials, and the public, Trump was characterized as deliberately upending democracy and working against the interests of the everyday person.

*Cluster 16: Anti-establishment.* Anti-establishment values framed this cluster's retweeting patterns. Bernie often campaigned on the message of promoting progressive policies to end the wealth gap in the US, but the grievances suggested by this cluster often promoted the complete redistribution of wealth, "I have a message for the top 1% and the large profitable corporations in this country: Enjoy the massive tax breaks and loopholes you have right now. Under a Bernie Sanders administration, we're going to end them and invest in working people."

*Cluster 17: Biden is a villain.* Villainizing one's opponent can also be a potent mobilization strategy if the source is perceived as credible [80] as character attacks rely upon well-worn tropes of heroes and villains [81]. This cluster tweeted with the conviction that Biden was, "not a decent person, a plagiarist, and pathological liar." Framing Biden as a villain offered this cluster the rhetorical maneuver to promote a Vote for Bernie. Unlike the progressive clusters, however, those from the Far Left deduced instead that if the Presidential election was a choice between Trump and Biden they would, ". . .never under any circumstances vote for Joe Biden. Ever."

*Cluster 18: M4A*. Across the ideological spectrum, many heralded Bernie's signature Medicare for All platform. However, instead of mirroring Bernie's call to action through unity, this cluster was shaped by anti-establishment thinking and conspiracy-adjacent perspectives by offering that even with a Democratic supermajority, the status quo democratic politician will, "…say no to it. What then? Keep waiting for a time that never arrives? You create your openings through force of will and public pressure. 300k Americans dead. Fight now." Similar to cluster 16, there is a collective "we" invoked that positioned Bernie supporters in opposition to the status quo and in conflict with the average Democrat. As they surmised, achieving healthcare for all may mean imposing one's will upon an unnamed entity.

## Linking framed values and social groups

Finally, we examined the intersection of collective identities and ideology, exploring how much identity and ideology aligned or diverged. We do so by looking at the use by each of the five Bernie-supporting social groups identified of the fourteen different framed value clusters identified in the who retweets what network As noted, we do so in the context of the prevalent assumption in social movement scholarship that conflates them as one and the same [3, 17]. We find that the five distinct social groups identified in the previous section vary in both the number and ideologies of the framed values expressed. Perhaps most obviously, Fig 5 shows that across all five social groups, popular tweets representing a moderate ideology were less prominent than those representing a progressive or far left ideology. The primary exceptions to this were in the "Get out the Vote" frame; over 20% of retweets in the Organizing for Bernie group, and over 15% of tweets by the Bernie Bro group did as well. This dearth of moderate ideologies aligns with the established idea that Bernie did not often appeal to a moderate base (see [17]).

Fig 5 also reveals three distinct ways in which identity and ideology functioned together to produce behavior on Twitter. First, in the case of the group of Young and/or Black voices, identity *drove* ideological expression, as represented by the values or issues framed. Nearly 70% of the retweets from these users expressed framed values that reflected a Young and/or Black identity. Second, in the case of the Popular Progressives and Green Party Socialists, ideology and identity functioned more equally in creating a movement-within-a-movement; ideologies expressed in retweet content mirrored the progressive and far-left identities put forth in user profile descriptions, respectively. Last, the Bernie Bro and Organizing for Bernie groups displayed cases where identity and ideology were not cleanly aligned and sometimes diverged. These groups used framed values across the ideological spectrum as a menu of options to encourage support for Bernie. Interestingly, it is precisely these clusters in which identity and ideology were not well aligned that users expressed tweets in support of Biden as a candidate once Bernie dropped out of the race. Whether or not this represents a meaningful distinction is, however, left to future work.

## Conclusion

Using a novel dataset and approach, we find distinctions in both who and what Bernie supporters retweeted that reflect systematic variations in the identities and ideologies of Bernie supporters. With respect to our methodology, we propose a new mixed-methods strategy to 1) identify distinct social groups within large, topical streams of Twitter data and 2) a means to further identify the framed values they exhibit and their associations with known ideologies. These findings challenge three prevailing themes in mainstream media and academic literature on social movements. First, while the Bernie Bro stereotype represents some Bernie supporters using Twitter, we provide convincing evidence that it is a misnomer when applied to everyone

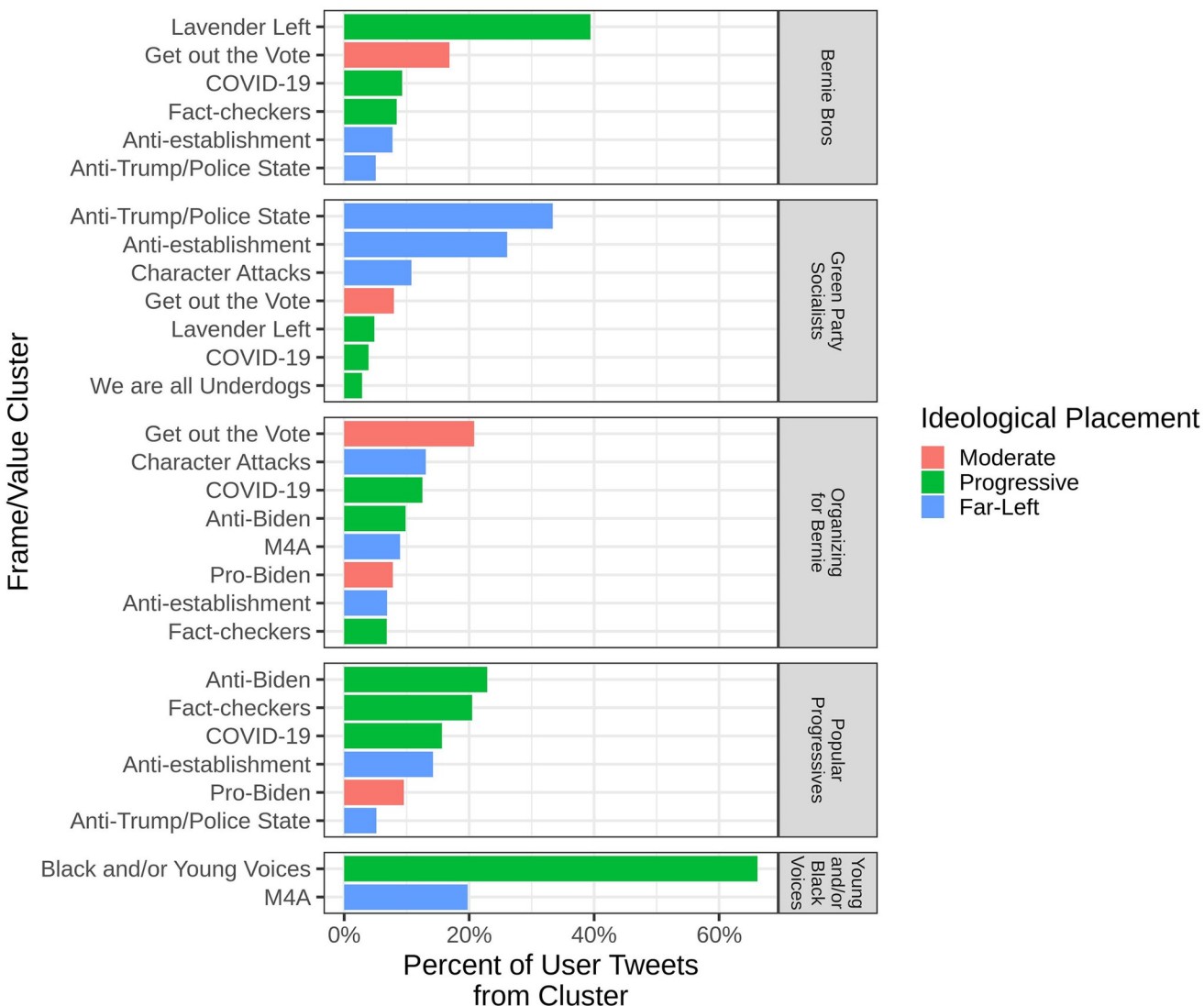

**Fig 5. The percentage of tweets (x-axis) sent by ordinary users in each of these five social groups (distinct subplots) that were assigned to each of the fourteen framed values clusters (y-axis) in the who retweets what network.** We show frames that make up at least 90% of the tweets in the given cluster of users.

who desired to see Bernie become the candidate for the Democratic Party during the 2020 election. Second, and related, we provide a case study that challenges assumptions of ideological and/or identity homogeneity within a social movement. Finally, our work emphasizes the utility of a careful analytical distinction between political identity and political ideology. In the context of social computing, our work provides a corrective on recently critiqued patterns in the field of an overemphasis on the study of the political right [47] and of dis- and misinformation relative to analyses of the broader ecosystem [82].

Critically, the present work in no way minimizes the lived experiences of those who experienced misogynistic and/or racist actions from Bernie supporters. Indeed, even though the present focuses on tweets containing election-relevant keywords, and thus is almost certain to miss content in tweet replies that are particularly virulent or hateful, we still observe some forms of misogyny in our data. Our work simply makes the case that a large number of Bernie

supporters had ideologies and identities that do not align with these kinds of actions or behaviors, or that are often the subjects of these kinds of actions. The perpetuation of the Bernie Bro stereotype, we believe, hinders these voices from the policies that these individuals advocate for. More generally, perpetuation of the Bernie Bros trope limits our knowledge of modern elections and variation—within identity and ideology—among supporters for the same candidate.

While our work offers several notable contributions, these should be considered in the context of several limitations to our study. First, our analysis does not validate that every account we identified was indeed a Bernie supporting account. It is thus possible that identity and/or ideological variation emerges from a mixing of accounts that were and were not Bernie supporting. However, given the variations that existed even among explicit Bernie supporters, it seems likely that at worst we overestimate such variation, rather than it disappearing entirely. Second, we do not make any attempt to remove bot accounts from our analysis. However, given that bots are not central to discussions around protests [83], an important component of online movement discussions, and that our analysis identifies Bernie-supporting accounts via an analysis of cohesive social groups in retweet networks, we believe our work may be fairly robust to corruption from bot accounts. Third, our qualitative analysis centers around output from a VSP model using a single setting of the parameter $k$. While we show in S1 Fig. 6 that the clusters identified are highly similar to those using other values of $k$ according to AMI, it is therefore possible that qualitative interpretation might vary if we were to adopt a different model.

Finally, while our work addresses known gaps in social movement scholarship about the importance of identity and ideological variation within the same movement, our study only focuses on one candidate; albeit along multiple dimensions of likely Bernie supporters. We suggest that to further parse ideology, identity, and framed value variation, future research might offer a comparative analysis of several candidates within the same party to further identify the mechanisms through which identity, ideology, and framed values cohere or conflict in social movement organizing. Indeed, as noted by other scholars, it is these kinds of intramovement disputes that can, "reveal a movement's weaknesses and provide opponents with a blueprint for launching attacks; rendering the movement vulnerable to divide and conquer tactics" ([8], pg. 696). Thus, it may be that Bernie was contending with not only other Democratic candidates, but himself, given the range of ideological and identity variation among the base of Bernie supporters that we identified. As such, our work points to what many Bernie supporters—especially progressives—may have perceived as a strength (i.e. a unity ticket, #NotMeUs) was a weakness to a broader audience of voters. The framed values that Bernie's platform offers, or those that resonate with the retweet network, may be spread too thin. Within this context, opponents in both the Democratic and Republican party (as well as in the popular press) can over-emphasize the specter of the Bernie Bro while downplaying moderate, progressive, and some far left supporters of Bernie. We leave it to future researchers to identify these dynamics.

## Supporting information

**S1 Fig. Mean AMI values for runs of VSP.** Mean AMI values for runs of VSP with 95% confidence intervals (y-axis) for each setting of $k$ for runs of VSP for the clustering of the who retweets whom (a) and who retweets what (b) analysis.
(ZIP)

**S1 File.**
(ZIP)

## Acknowledgments

The authors wish to thank Zijian An for assistance in preparing the initial dataset and the anonymous reviewers and editor for their thoughtful comments.

## Author Contributions

**Conceptualization:** Stef M. Shuster, Celeste Campos-Castillo, Kenneth Joseph.

**Data curation:** Stef M. Shuster, Kenneth Joseph.

**Formal analysis:** Stef M. Shuster.

**Investigation:** Celeste Campos-Castillo, Navid Madani.

**Methodology:** Stef M. Shuster, Celeste Campos-Castillo, Navid Madani, Kenneth Joseph.

**Software:** Navid Madani.

**Writing – original draft:** Stef M. Shuster, Kenneth Joseph.

**Writing – review & editing:** Stef M. Shuster, Celeste Campos-Castillo, Kenneth Joseph.

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
