## [Decision Letter · Decision Letter 0]

21 Jun 2023

PONE-D-23-10475Who Supports Bernie? Analyzing Identity and Ideological Variation on Twitter During the 2020 Democratic PrimariesPLOS ONE

Dear Dr. shuster,

Thank you for submitting your manuscript to PLOS ONE. After careful consideration, we feel that it has merit but does not fully meet PLOS ONE’s publication criteria as it currently stands. Therefore, we invite you to submit a revised version of the manuscript that addresses the points raised during the review process.

We look forward to receiving your revised manuscript.

Kind regards,

Francesco Branda

Academic Editor

PLOS ONE

Journal Requirements:

2. PLOS ONE has specific criteria regarding data-sharing and availability (https://journals.plos.org/plosone/s/data-availability). Specifically, these guidelines require that authors to make all data necessary to replicate their study’s findings publicly available without restriction at the time of publication. When specific legal or ethical restrictions prohibit public sharing of a data set, authors must indicate how others may obtain access to the data. To that effect, please clarify in your data availability statement whether the data for your study can be made available at time of publication, if accepted.

Also, in your Methods section, please include additional information about your dataset and ensure that you have included a statement specifying whether the collection and analysis method complied with the terms and conditions for the source of the data.

"KJ and NM are supported by a grant from the Office of Naval Research (https://www.nre.navy.mil/) MURI ONR # N00014-20-S-F003 and the National Science Foundation (https://nsf.gov/) IIS-2145051."

"I have read the journal's policy and the authors of this manuscript have the following competing interests: ss, NM, CC,  KJ"

Reviewers' comments:

Reviewer's Responses to Questions

**Comments to the Author**

1. Is the manuscript technically sound, and do the data support the conclusions?

Reviewer #1: Yes

Reviewer #2: Yes

Reviewer #3: Yes

2. Has the statistical analysis been performed appropriately and rigorously? 

Reviewer #1: Yes

Reviewer #2: I Don't Know

Reviewer #3: Yes

3. Have the authors made all data underlying the findings in their manuscript fully available?

Reviewer #1: No

Reviewer #2: No

Reviewer #3: Yes

4. Is the manuscript presented in an intelligible fashion and written in standard English?

Reviewer #1: Yes

Reviewer #2: Yes

Reviewer #3: Yes

5. Review Comments to the Author

Reviewer #1: This paper contributes a substantial novel social media dataset related to Bernie Sanders’ 2020 presidential campaign, contributes an interesting method of identifying identity groups through retweet networks, and provides qualitative analysis of the identified clusters. Using mixed methods, the clusters of users are labeled as Bernie supporting or not, and coded for ideology. With this, the authors claim that online Bernie supporters are composed of diverse identity and ideological groups. I found the claims to be well-supported by the presented analysis.

There is little information about the criteria used to include keywords. The authors should consider expanding on their choices and how they balanced the tradeoff between precision and recall when including keywords, and how their choices were likely to effect the dataset.

The authors might consider performing topic modeling, using either BERTopic or LDA. The results section is quite extensive, which may be unavoidable due to the mixed methods being employed, but some kind of quantitative summarization of the identified clusters could be helpful. Using sentence embeddings, with tweets colored by user cluster could help illustrate some textual relationships for readers. https://www.sbert.net/

The authors clearly communicate their process for the mixed methods, but have not provided links to either their code to reproduce their quantitative analysis nor a repository with tweet ids or user_ids with cluster labels.

Typo on line 148, with perhaps more.

Reviewer #2: I am not sure whether there is an actual methodological contribution, given that the authors are replicating the approach presented in ref'17; if I got it wrong, please try to underline the novelty of your approach.

I think the Background section is a bit too long and could be shortened and made more concise; I also feel the manuscript relies too much on social science background for PLOS One readership -- the same applies for some arguments in the Results/Discussion section. I think the authors could unwind a bit the sociological implications as they might be suitable for a more specialized journal that this one.

The choice of parameters in different filtering phases of the Methods section could be motivated a bit.

If it's not time consuming I would try a robustness analysis changing the number of clusters around the optimal value to see how results would be affected -- it could strengthen the conclusions.

What does "factor loadings" mean at line 333?

Please motivate why you choose to analyze only 14 clusters (line 372).

Is there overlap between Bernie Sanders and other primary candidates? Might be worth mentioning this.

I think Cluster 3 is missing from Figure 2 (line 401).

Please specify what influencers are (line 421).

I am bit confused by the description of clusters: first only 5 clusters are mentioned, but then many more appear (I think it has to do with the tweet-based clustering). Please try to clarify this distinction throughout the Results section.

Figure 1 could be improved by adding a legend for vertical lines in the text; I would also suggest using a solid line with dashed areas for a moving average with confidence intervals, and unconnected dots for the daily observations.

Figure 2: are all numbers really necessary? Might be useful to highlight only clusters that are mentioned and described in the text.

Figure 3: I would separate the normal users from influencers and probably use a log scale to show numbers for the red clusters.

Figure 5: I feel like this figure contains too much information, but I can't think of a better way to visualize it.

Please add a "Data Availability" statement, mentioning potential limitations in accessing Twitter APIs nowadays.

Reviewer #3: Title: "Who Supports Bernie? Analyzing Identity and Ideological Variation on Twitter During the 2020 Democratic Primaries”

The paper titled "Who Supports Bernie? Analyzing Identity and Ideological Variation on Twitter During the 2020 Democratic Primaries” is a commendable and insightful piece of scholarly work. This paper presents an astute exploration of an under-examined aspect of modern political science. It not only enriches our understanding of the 2020 Democratic Primaries and their social media dynamics, but it also offers a rigorous and innovative approach to the analysis of Twitter data.

The authors have skillfully leveraged a large dataset, gathering millions of tweets related to the Bernie Sanders campaign, to build a picture of his supporters. The paper offers a comprehensive, nuanced exploration of Bernie Sanders supporters' demographic and ideological profiles. This investigation is timely and pertinent and lends an empirical underpinning to the swirling discussions about the Sanders coalition.

A key strength of the paper is the adept use of sparse Principal Component Analysis (PCA) for identifying the underlying ideological dimensions among Bernie Sanders supporters. Applying machine learning techniques to analysing social media data is commendable and elevates the work's overall sophistication.

Nevertheless, one of the main remarks is the paper's presumption of linear assumptions inherent in the sparse PCA method. Sparse PCA assumes that the data is linear and normally distributed, which might not be the case with complex socio-political discourses and identities embedded within Twitter data. While this method offers many benefits, such as effective dimension reduction and interpretability, its assumptions may not entirely align with the data's inherent structure. Moreover, and this is a curiosity more than demand, what about the sparsity level? The most significant limitation of Sparse PCA is deciding how sparse each principal component should be. Unlike standard PCA, which does not require such a decision, Sparse PCA requires an additional user-defined parameter to control the sparsity level. This can make the results of Sparse PCA more sensitive to this choice, and it may not always be obvious what the best sparsity level is.

The last question is about the issue of scaling. It might not concern your analysis, but please state so. Just like standard PCA, Sparse PCA can be sensitive to the scaling of the input variables. However, this issue can become more pronounced with Sparse PCA, as the introduction of sparsity can make the method even more sensitive to the relative scales of the variables.

In this context, I would suggest the authors provide a more in-depth justification for selecting the sparse PCA over alternative methods, such as the latent profile analysis. The latent profile analysis is a multivariate method that identifies hidden groups within the dataset, which may present a better fit for the ideological and identity variations in question. A brief comparative discussion of these methods would be enlightening and serve to further underscore the study's methodological rigour.

Overall, this paper is valuable to the scholarship around political behaviour, social media's role in politics, and particularly the fascinating Sanders phenomenon. With some minor clarifications and justification on the choice of sparse PCA, this paper could serve as an important reference for future studies examining the link between political ideology, identity, and social media discourse.

Overall, this paper is a valuable addition to the scholarship around political behavior, social media's role in politics, and in particular, the fascinating Sanders phenomenon. With some minor clarifications and justification on the choice of sparse PCA, this paper could serve as a leading reference for future studies examining the link between political ideology, identity, and social media discourse. The authors are to be commended for this significant contribution to the field.

6. PLOS authors have the option to publish the peer review history of their article (what does this mean?). If published, this will include your full peer review and any attached files.

Reviewer #1: No

Reviewer #2: No

Reviewer #3: No

---

## [Author Response · Author response to Decision Letter 0]

28 Sep 2023

Response has been included in the submission, uploaded as a "Response to Reviewers"

---

## [Decision Letter · Decision Letter 1]

8 Nov 2023

Who Supports Bernie? Analyzing Identity and Ideological Variation on Twitter During the 2020 Democratic Primaries

PONE-D-23-10475R1

Dear Dr. shuster,

We’re pleased to inform you that your manuscript has been judged scientifically suitable for publication and will be formally accepted for publication once it meets all outstanding technical requirements.

Kind regards,

Francesco Branda, Ph.D.

Academic Editor

PLOS ONE

Additional Editor Comments (optional):

Reviewers' comments:

Reviewer's Responses to Questions

**Comments to the Author**

1. If the authors have adequately addressed your comments raised in a previous round of review and you feel that this manuscript is now acceptable for publication, you may indicate that here to bypass the “Comments to the Author” section, enter your conflict of interest statement in the “Confidential to Editor” section, and submit your "Accept" recommendation.

Reviewer #1: All comments have been addressed

Reviewer #2: All comments have been addressed

2. Is the manuscript technically sound, and do the data support the conclusions?

Reviewer #1: Yes

Reviewer #2: Partly

3. Has the statistical analysis been performed appropriately and rigorously? 

Reviewer #1: Yes

Reviewer #2: I Don't Know

4. Have the authors made all data underlying the findings in their manuscript fully available?

Reviewer #1: Yes

Reviewer #2: Yes

5. Is the manuscript presented in an intelligible fashion and written in standard English?

Reviewer #1: Yes

Reviewer #2: Yes

6. Review Comments to the Author

Reviewer #1: (No Response)

Reviewer #2: I think the authors have addressed most concerns raised by reviewers and provided reasonable responses when they did not satisfied specific requests.

7. PLOS authors have the option to publish the peer review history of their article (what does this mean?). If published, this will include your full peer review and any attached files.

Reviewer #1: No

Reviewer #2: No

---

## [Editor Report · Acceptance letter]

10 Nov 2023

PONE-D-23-10475R1 

Who Supports Bernie? Analyzing Identity and Ideological Variation on Twitter During the 2020 Democratic Primaries 

Dear Dr. Shuster:

I'm pleased to inform you that your manuscript has been deemed suitable for publication in PLOS ONE. Congratulations! Your manuscript is now with our production department. 

Kind regards, 

on behalf of

Dr. Francesco Branda 

Academic Editor

PLOS ONE